# MEDIMP: 3D Medical Images with clinical Prompts from limited tabular data for renal transplantation

**Leo Milecki**[1](✉)                                              LEO.MILECKI@CENTRALESUPELEC.FR
[1] *MICS, CentraleSupelec, Paris-Saclay University, Inria Saclay, France*

**Vicky Kalogeiton**[2]                                      VICKY.KALOGEITON@POLYTECHNIQUE.EDU
[2] *LIX, École Polytechnique, CNRS, Institut Polytechnique de Paris, France*

**Sylvain Bodard**[3,6]                                            SYLVAIN.BODARD@APHP.FR
[3] *Department of Adult Radiology, Necker Hospital, Paris University, France*

**Dany Anglicheau**[4,6]                                         DANY.ANGLICHEAU@APHP.FR
[4] *Department of Nephrology and Kidney Transplantation, Necker Hospital, APHP, France*

**Jean-Michel Correas**[3,6]                                   JEAN-MICHEL.CORREAS@APHP.FR

**Marc-Olivier Timsit**[5,6]                                   MARC-OLIVIER.TIMSIT@APHP.FR
[5] *Department of Urology, HEGP, Necker Hospital, Paris University, France*
[6] *UFR Médecine, Paris-Cité University, France*

**Maria Vakalopoulou**[1]                                MARIA.VAKALOPOULOU@CENTRALESUPELEC.FR

**Editors:** Accepted for publication at MIDL 2023

## Abstract

Renal transplantation emerges as the most effective solution for end-stage renal disease. Occurring from complex causes, a substantial risk of transplant chronic dysfunction persists and may lead to graft loss. Medical imaging plays a substantial role in renal transplant monitoring in clinical practice. However, graft supervision is multi-disciplinary, notably joining nephrology, urology, and radiology, while identifying robust biomarkers from such high-dimensional and complex data for prognosis is challenging. In this work, taking inspiration from the recent success of Large Language Models (LLMs), we propose MEDIMP – Medical Images with clinical Prompts – a model to learn meaningful multi-modal representations of renal transplant Dynamic Contrast-Enhanced Magnetic Resonance Imaging (DCE MRI) by incorporating structural clinicobiological data after translating them into text prompts. MEDIMP is based on contrastive learning from joint text-image paired embeddings to perform this challenging task. Moreover, we propose a framework that generates medical prompts using automatic textual data augmentations from LLMs. Our goal is to learn meaningful manifolds of renal transplant DCE MRI, interesting for the prognosis of the transplant or patient status (2, 3, and 4 years after the transplant), fully exploiting the limited available multi-modal data most efficiently. Extensive experiments and comparisons with other renal transplant representation learning methods with limited data prove the effectiveness of MEDIMP in a relevant clinical setting, giving new directions toward medical prompts. Our code is available at https://github.com/leomlck/MEDIMP.

**Keywords:** Contrastive Learning, Natural Language Processing, LLM, MRI, Renal transplantation, Medical Prompts.

## 1. Introduction

End-stage renal disease is characterized by an irremediable reduction in kidney function, and renal replacement therapy is required to save the patient's life. Being more cost-effective than long-term dialysis and highly improving quality of life, renal transplantation emerges as the most effective solution (Suthanthiran and Strom, 1994). However, a substantial risk of transplant chronic dysfunction persists and may lead to graft loss or patient death (Hariharan et al., 2021). In clinical practice, the graft health status is primarily indicated by calculating the glomerular filtration rate (GFR) from the creatinine level resulting from blood tests. Medical imaging plays a substantial role in further examinations, and diverse imaging modalities have been investigated to monitor renal transplants (Sharfuddin, 2013).

Learning powerful representations of medical imaging is of utmost importance, given the usual small size and limited annotations available. In such a setting, learning is performed in two stages. In the first stage, different self-supervised or weakly-supervised learning methods (Taleb et al., 2020; Krishnan et al., 2022) are used on the available imaging datasets, applying different types of learning, such as contrastive or adversarial learning (Sowrirajan et al., 2021; Azizi et al., 2021; Boyd et al., 2021). Such representations are then frozen or fine-tuned for different downstream tasks, for which the amount of information is insufficient for fully supervised learning. Such pretrainings could provide better representations and outperform ImageNet pretrained networks when applied to medical imaging. However, they may produce suboptimal representations for the downstream tasks that merely capture spurious correlations (Arjovsky et al., 2019). More particularly, for renal transplantation, Milecki et al. (2022) proposed weakly-supervised tasks from clinical information to learn rich representations of Dynamic Contrast-Enhanced Magnetic Resonance Imaging (DCE MRI) using a single continuous attribute, confirming that combining imaging and clinical data leads to powerful biomarkers for prognosis.

Recent advances in Natural Language Processing (NLP) make textual data a potent candidate for designing weakly-supervised tasks to train computer vision models. Multiview contrastive learning (Bachman et al., 2019) has been investigated to take advantage of jointly training an image and text encoder (Zhang et al., 2020; Radford et al., 2021; Jia et al., 2021; Müller et al., 2022). For natural images, Radford et al. (2021) produced robust representations using 400 million (image, text) pairs, reporting competitive results on several downstream tasks on unseen datasets compared to fully supervised baselines. Zhang et al. (2020) used chest X-rays and pathology descriptions from radiology experts' diagnoses. Müller et al. (2022) extended this joint image-text representation learning for localized tasks like semantic segmentation or object detection. All these studies consider 2D images and the medical ones used the MIMIC-CXR database, the largest dataset containing paired medical images and radiology reports. However, such data curation is arduous and highly time-consuming for medical experts. Moreover, such reports mainly contain information about the corresponding imaging exam and do not focus on other comorbidities.

Therefore, we propose to go one more step forward by generating representations using paired imaging and clinicobiological attributes in a relevant clinical setup with limited data. We explore recent NLP advances in Large Language Models (LLMs). In particular, ChatGPT (OpenAI, 2022), a 175B parameters model, offers a powerful tool to produce textual data. Specifically, textual data allow advantages as opposed to tabular data for medical ap-

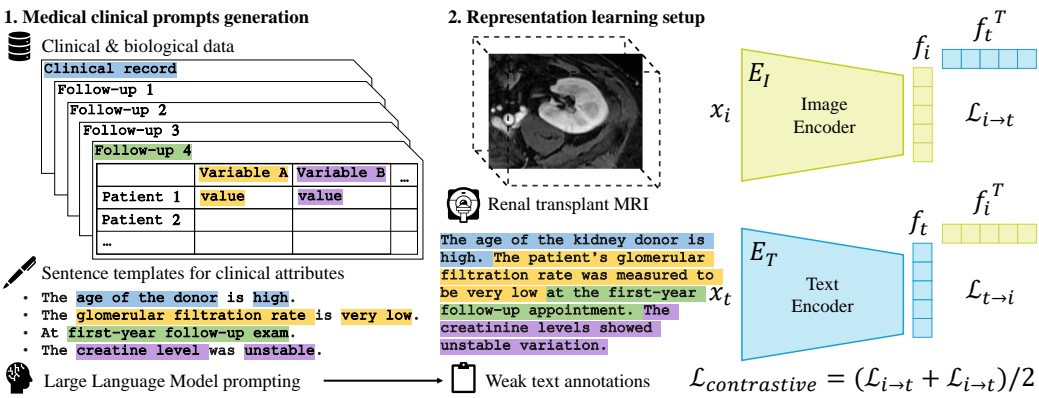

Figure 1: **Overview of our method MEDIMP – Medical Images with clinical Prompts.**
1. Medical prompts are generated from clinicobiological data using predefined templates
of sentences, given as inputs to Large Language Models to produce augmented text data.
2. The medical prompts are used to learn multi-modal representations of renal transplants
DCE MRI using contrastive learning from image-text pairs.

plications: (1) Contextual information: Textual data contains rich contextual information,
helping to understand the underlying patterns in the data better; (2) Better representation:
text can provide a more expressive representation of the information contained in the data,
leading to improved performances as our model better capture the complexities of the clin-
icobiological data. (3) Transferability and Interpretability: text is often more transferable
across domains than tabular data. Moreover, text is more interpretable by humans, which
is valuable for validating and understanding the decisions made by the proposed method.

In this work, we introduce MEDIMP (**MED**ical **IM**ages with clinical **P**rompts). This
method learns relevant DCE MRI representations of renal transplants using contrastive
learning from pairs of 3D images and clinicobiological prompts. The learned manifold
enabled us to outperform state-of-the-art methods in the challenging task of kidney function
prediction 2, 3, and 4 years post-transplantation from 4 DCE MRI follow-up exams. Our
contributions are: (i) We propose a semi-automatic medical prompt generation from tabular
data; to the best of our knowledge, this is the first work to propose such an approach for
augmenting medical textual data; (ii) We extend existing approaches that combine text
and imaging data by integrating 3D medical inputs and fine-tuning strategies; our approach
allows using pretrained NLP models on a limited amount of textual data.

## 2. Method

Our multi-modal representations are based on contrastive learning, coupling imaging, and
text embeddings. Our text relies on attributes that are easily accessible, widely used in
clinical practice, and supplementary to imaging data. Our goal is to use the learned manifold
of renal transplant DCE MRI for the prognosis of the transplant or patient status after $2, 3$,
and 4 years post-transplantation. An overview of the method is presented in Figure 1.

### 2.1. Contrastive learning from joint text-image pairs

The first component of MEDIMP is pretraining an image encoder $E_\mathrm{I}$, and a text encoder $E_\mathrm{T}$, following a contrastive learning scheme using image-text pairs. Let us denote $(x_i, x_t) \in \mathbb{R}^{B \times N_x \times N_y \times N_z} \times \mathbb{R}^{B \times T}$ a batch of $B$ corresponding pairs of an 3D MRI volume $x_{ib}$ and a tokenized text $x_{tb}$ for $b \in [\![1, B]\!]$. Both encoders transform $x_i$ and $x_t$, in $f_i = E_\mathrm{I}(x_i)$ and $f_t = E_\mathrm{T}(x_t)$ respectively, to a batch of $D$ dimensional embeddings. Both encoders are jointly trained to maximize the cosine similarity between the $B$ pairs of image and text embeddings by optimizing the two following losses:

$$\mathcal{L}_{i \to t} = \sum_{b=1}^{B} - \log \frac{e^{\cos(f_{ib}, f_{tb})/\tau}}{\sum_{k=1}^{B} e^{\cos(f_{ib}, f_{tk})/\tau}} \quad , \tag{1}$$

where $\cos(\cdot, \cdot)$ is the cosine similarity function and $\tau \in \mathbb{R}^+$ a learned temperature parameter. Such loss was first proposed as the InfoNCE loss (van den Oord et al., 2018) to maximize a lower bound on mutual information and is widely used in recent uni-modal contrastive learning frameworks (Chen et al., 2020). $\mathcal{L}_{i \to t}$ enforces the image embeddings to align to the text embeddings and is, therefore, asymmetric. Similarly, we define $\mathcal{L}_{t \to i}$:

$$\mathcal{L}_{t \to i} = \sum_{b=1}^{B} - \log \frac{e^{\cos(f_{tb}, f_{ib})/\tau}}{\sum_{k=1}^{B} e^{\cos(f_{tb}, f_{ik})/\tau}} \quad . \tag{2}$$

The total loss is obtained by averaging Equation (1) and Equation (2), denoted as $\mathcal{L}_{\mathrm{contrastive}}$. $\mathcal{L}_{\mathrm{contrastive}}$ learns a multi-modal feature space by jointly optimizing $E_\mathrm{I}$ and $E_\mathrm{T}$ to maximize the cosine similarity of the embeddings $f_i$ and $f_t$ between the $B$ true pairs per batch and minimizing the cosine similarity of the $B^2 - B$ false pairs.

### 2.2. Medical prompts from structural clinicobiological data

To exploit image-text pairing with contrastive learning, as well as the expression and encoding capability of recent NLP model advances, such as LLMs (Brown et al., 2020; OpenAI, 2022; Radford et al., 2021). We propose a framework to generate textual data from structural clinicobiological data that describe variables used in clinical practice and linked to the graft survival. The process is displayed on the left side of Figure 1. First, medical experts guided us to set thresholds to categorize continuous variables into text labels such as "low", "high", "stable", and "unstable" and to produce one *template sentence* per variable of interest, e.g., "the GFR of the patient is very low at the first-year follow-up exam". However, templates offer only one way of expressing the information of the variables. Indeed, the richness of language vocabulary can provide a variety of descriptions for the same information, such as "During the first-year follow-up visit, the transplant patient's GFR is found to be very low.", or "The transplant patient's GFR is assessed as very low at the date follow-up examination.", thus generating descriptive text to train the proposed contrastive scheme. This richness was leveraged by recent advances in LLMs at training; hence, they offer robust NLP tools. Specifically, we use the dialogue LLM ChatGPT (OpenAI, 2022) to produce $N = 10$ textual data augmentations for each template sentence. All generated prompts are reported in Appendix A.

## 2.3. Implementation details

The image encoder followed a 3D ResNet50 architecture initialized with CLIP (Radford et al., 2021) weights, a model pretrained on 400 million (image, text) pairs collected from the internet. We extended the attention-based pooling layer of CLIP to 3D and duplicated the weights to 3D in depth to match our data dimension. For the text encoder, we used the BERT (Devlin et al., 2019) architecture initialized with the Bio+Clinical BERT (Alsentzer et al., 2019) model pretrained on the MIMIC clinical notes (Johnson et al., 2016). The first 11 layers of Bio+Clinical BERT were frozen, fine-tuning the last layer of the transformer with our contrastive task. Appendix B summarises the ablation of fine-tuning more layers for our task. Appendix C lists image augmentations and training hyperparameters.

## 3. Data

Our study was approved by the Institutional Review Board, which waived the need for patients' consent. The data cohort corresponds to study reference ID-RCB: 2012-A01070-43 and ClinicalTrials.gov identifier: NCT02201537. The data used in this study are anonymized. Our imaging data are based on DCE MRI series collected from 105 subjects (split as 72/5 training/validation, and 28 test). Each subject underwent up to 4 follow-up exams, taking place approximately 15 days, 30 days, 3 months, and 1 year post-transplantation. DCE MRI volumes preprocessing is described in Appendix C. To provide the clinicobiological data used to generate text annotations and the endpoints, all 77 patients in the train set were regularly subjected to blood tests before the transplantation and several years after to measure the serum creatinine (Creat) level in $\mu mol.L^{-1}$. The donor's age variable and the GFR value at each follow-up exam were also collected. For the 28 test subjects, these clinicobiological attributes were not accessible during this study. Resulting from blood tests, Creat is a primary indicator of kidney function used in clinical practice. The binary classification downstream task is obtained when binarizing the Creat value using a threshold of 110 $\mu mol.L^{-1}$ at different prediction dates. The Creat target prediction value is calculated as the mean over three months before and after the prediction dates.

## 4. Experiments & Results

First, we visualize the representations of DCE MRI from the trained image encoder using t-SNE decomposition (van der Maaten and Hinton, 2008). Figure 2 shows the different reduced feature spaces obtained by the model trained on all available clinicobiological variables ($n_{cl} = 4$), projected specifically and adding colormaps for each attribute. We evaluate the clustering of the imaging features toward the clinicobiological information covered by our weak text annotations, adding augmented images to check the tendency better. While the continuous variables were categorized and transformed to medical prompts, MEDIMP image encoder demonstrates relevant representations towards (B) the GFR and (C) the Creat. The t-SNE decomposition does not reveal favorable representations regarding (D) Donor's Age. On the contrary, the obtained feature space serves well the (A) Exam date information, where we retrieve better clustered very early exams (D15, red) and late exams (M12, blue) due to their respective distance to the transplantation surgery.

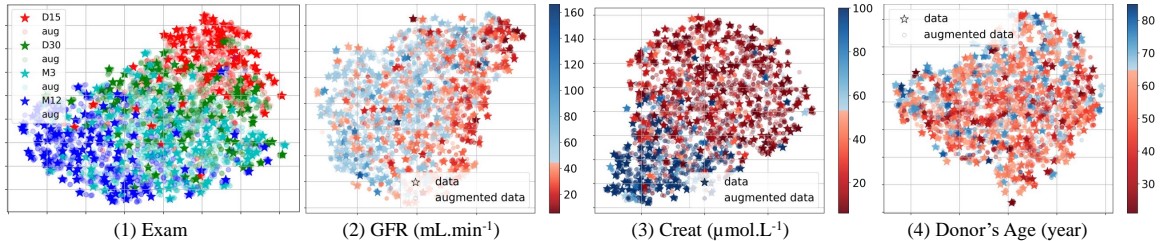

|  |  |  |  |
|---|---|---|---|
| (1) Exam | (2) GFR (mL.min⁻¹) | (3) Creat (µmol.L⁻¹) | (4) Donor's Age (year) |

Figure 2: **t-SNE visualizations of the features of the last layer of MEDIMP image encoder using the DCE MRI exams.** Colormaps are set by the 4 variables of interest value used for the medical prompts: (1) Exam (exam date), (2) GFR $(mL.min^{-1})$, (3) Creat $(\mu mol.L^{-1})$, and (4) donor's age (year). Stars symbol display the real data while circles the augmented (aug) data. D15, D30, M3, and M12 are the four exam timestamps.

**Downstream task and metrics.** We evaluate MEDIMP on the downstream task of serum creatinine (Creat) prediction from the imaging features of 4 follow-up exams using a light transformer architecture tailored for missing follow-up exams, proposed by Milecki et al. (2022). Following the authors' evaluation, 10-fold cross-validation was performed on the training set, and results for the main models are summarised in Appendix D. To make the task more challenging, we evaluate the performance of the representations at 2, 3, and 4 years post-transplantation and also report the mean over the three predictions for the 28 test subjects. The two evaluation metrics used were the F1 score and the area under the receiver operating characteristic curve (AUC).

**Ablation.** First, we ablate all information used in our method MEDIMP by adding different combinations of the clinicobiological measurements in the medical prompts, i.e., the glomerular filtration rate (GFR) at the exam date, the timestamp of the patient's exam (Exam), the creatinine levels variation from the previous exam (Creat), and the age of the donor (D.A.). We report the results in the bottom part of Table 1 (MEDIMP). We observe that the best mean scores over the three predictions are obtained using all the available medical prompts (last row). Moreover, the AUC decreases over the prediction date, showing the increasing prognosis difficulty with time. Using only the GFR prompts (row 8), AUC scores are just above random, indicating the need for more descriptive text.

**Comparison to the state of the art.** Table 1 reports the results when comparing the proposed MEDIMP with the previous state-of-the-art for this task (Milecki et al., 2022), denoted as CosEmbLoss (row 2 & 3). Note that the main model of CosEmbLoss uses only GFR information. Hence, it is directly comparable to MEDIMP when using only GFR (8th row). These experiments reveal that smaller and more compact models, such as the CosEmbLoss perform better than big models when the text information is not very rich. However, when more variables are integrated, our proposed methods outperform the CosEmbLoss. For a fair comparison, we also compare MEDIMP to several baselines with the same level of information. In particular, we evaluate against four baselines, denoted as the CosEmbLoss++, where we gradually add the same level of information as in MEDIMP. In practice, we optimize the same two-stream approach by averaging several cosine embedding losses

Table 1: **Comparison of MEDIMP vs SOTA.** We evaluate the performance at 2,3,4 years post-transplantation and report the mean. Ablations in weak annotations from either the comparison CosEmbLoss pretaining or our proposed generated textual data are denoted as GFR (GFR at exam), Exam (which follow-up), Creat (creatinine variations from the previous exam), and D.A. (the donor's age). We report F1 score (F1), and ROC AUC (AUC). **Bold**, Underlined indicates the top **1**, 2 performing combinations, respectively.

| Method | Weak annotations | | | | 2 years | | 3 years | | 4 years | | Mean | |
|---|---|---|---|---|---|---|---|---|---|---|---|---|
| | GFR | Exam | Creat | D.A. | AUC | F1 | AUC | F1 | AUC | F1 | AUC | F1 |
| CLIP weights | | | | | 62.6 | 73.7 | 52.5 | 78.1 | 51.3 | 54.6 | 55.5 | 68.8 |
| CosEmbLoss | ✓ | | | | 76.2 | 86.4 | 77.8 | 70.6 | 67.0 | 77.3 | 73.6 | 78.1 |
| CosEmbLoss | | | | ✓ | 75.5 | 81.1 | 75.6 | 68.8 | 66.1 | 78.1 | 72.4 | 76.0 |
| CosEmbLoss++ | ✓ | | | ✓ | 84.4 | 88.9 | 82.5 | **86.4** | 73.9 | 85.7 | 80.3 | 87.0 |
| CosEmbLoss++ | ✓ | | ✓ | | 81.6 | 87.8 | 71.3 | 85.1 | 71.3 | 90.2 | 74.7 | 87.7 |
| CosEmbLoss++ | ✓ | | ✓ | ✓ | 78.2 | 87.0 | 75.0 | 83.3 | 74.8 | 87.0 | 76.0 | 85.8 |
| CosEmbLoss++ | ✓ | ✓ | ✓ | ✓ | 75.5 | 85.7 | 62.0 | 69.8 | 63.5 | 80.9 | 67.0 | 78.8 |
| MEDIMP | ✓ | | | | 56.5 | 83.3 | 51.9 | 79.1 | 49.6 | **90.2** | 52.6 | 84.2 |
| MEDIMP | ✓ | ✓ | | | 81.0 | **89.4** | 81.9 | 80.0 | 74.8 | 84.4 | 79.2 | 84.6 |
| MEDIMP | ✓ | ✓ | | ✓ | 76.9 | 73.2 | **86.3** | 85.7 | 74.8 | **90.2** | 79.3 | 83.0 |
| MEDIMP | ✓ | ✓ | ✓ | | 72.8 | 86.4 | 71.9 | 81.0 | 71.3 | 71.8 | 72.0 | 79.7 |
| MEDIMP | ✓ | ✓ | ✓ | ✓ | **85.0** | **89.4** | 84.4 | 83.7 | **75.7** | **90.2** | **81.7** | **87.8** |

based on the number of variables of interest incorporated. We report these results in rows 4-7 of Table 1. We observe that CosEmbLoss++ achieves its best performance when using a combination of 2 variables. The best mean AUC is 80.3% with GFR and D.A., and the best F1 is 87.7% with GFR and Creat, which are lower than the best MEDIMP performances. Note, the Exam information was only added to one CosEmbLoss++ combination (7th row) as this variable is less adapted to CosEmbLoss approach. Nevertheless, combining the 3 variables of interest with CosEmbLoss++ lowers the performance to 76.0% AUC and 85.8% F1 in Mean. Overall, MEDIMP with all medical prompts results in the best predictions at 2 and 4 years post-transplantation.

**Medical Prompt generation.** To demonstrate the relevance of the proposed approach for medical prompt generation, we compare our main model with two other approaches that produce text information. The first one is noted as "Manual" and comprises all the templates indicated by the medical experts, corresponding to only one sentence per variable of interest. Note that this is the base of our proposed medical prompting without using the prompt expansion method described in Section 2.2. The second one uses an existing NLP model, T5 (Raffel et al., 2020), to produce sentences from structural data. For a fair comparison, we train the T5 model on the WebNLG 2020 data (Gardent et al., 2017) and infer it on our data to generate text, denoted as "T5 WebNLG". The results are summarised in Table 2, highlighting the superiority of our method. The "T5 WebNLG" approach offers a competitive F1 for all the predictions, although the AUC is the lowest except for the 2 years prediction. We show in Appendix E examples of generated texts from these three approaches. "Manual" approach lacks diversity in the text data, and therefore the training process of our proposed approach without text augmentations is more challenging.

Table 2: **Quantitative evaluation of the proposed method against other text generation methods.** All medical prompts were used. We report F1 score (F1), and ROC AUC (AUC). **Bold**, Underlined indicates the top **1**, 2 performing combinations, respectively.

| Method | 2 years | | 3 years | | 4 years | | Mean | |
|---|---|---|---|---|---|---|---|---|
| | AUC | F1 | AUC | F1 | AUC | F1 | AUC | F1 |
| MEDIMP | **85.0** | **89.4** | **84.4** | **83.7** | 75.7 | **90.2** | **81.7** | **87.8** |
| Manual | 74.2 | 76.2 | 80.6 | 62.1 | **80.0** | 76.9 | 78.3 | 71.7 |
| T5 WebNLG | 74.8 | 85.7 | 78.8 | 83.3 | 74.8 | 85.7 | 76.1 | 84.9 |

## 5. Discussion & Conclusion

Our experiments have shown improvements in the representation learning capabilities of deep image encoders for renal transplantation MRI compared to the previous state-of-the-art approach for the specific application of renal transplant function forecasting. MEDIMP aimed at enhancing representation learning approaches using external data, leveraging the power of deep NLP models, such as LLMs, and introducing a novel process to incorporate relevant clinicobiological medical information. We deem that such an approach crossing modalities in medical research would highly improve the capacity to understand complex biological and medical phenomena. However, some limitations of our framework remain to be analyzed. (1) First, although limited data is part of the challenges of this study, supplementary test data would indubitably support validating our method. To our knowledge, no public medical imaging dataset offers simultaneously longitudinal imaging, biological, as well as clinical data for each patient for prognosis at different times. Nevertheless, our proposed framework could be easily translated to similar datasets having imaging data and any type of tabular data. (2) Second, this work constitutes a first attempt to generate medical prompts as text information from a few clinicobiological variables of interest, which are crucial to apprehend complex medical concepts. Thus, we would seek to extend this work to exploit more variables to guide the training of the image encoders. Using our proposed framework this can be easily implemented since only a few templates will need to be defined to automatically generate text augmentations. (3) Finally, the study of the development and comprehension of LLMs is recent. We expect such NLP models to increase the emphasis on using textual data. However, the main drawback of these models is their reliability, in the sense that inaccurate generation might be challenging to detect. In this work, we use ChatGPT to enable relevant and robust text augmentations. No information from the sensitive medical data was leaked into the prompts, which is an essential process in the medical domain. One could think of several ways to further use such models as powerful tools, for instance, through prompt engineering tuned for medical cases.

We have introduced MEDIMP, an approach to learn powerful manifolds of renal transplant DCE MRI data toward transplant function forecasting in the context of renal transplantation monitoring. This approach provides an elegant way to incorporate clinical or biological information into the learning process of feature extraction of medical imaging data. The presented representation learning strategy enabled us to outperform the state of the art in the challenging task of creatinine prediction. These promising results advocate using textual data from emerging LLMs to assist in training robust medical imaging models.

## Acknowledgments

This work was performed using HPC resources from the "Mésocentre" computing center of CentraleSupelec and École Normale Supérieure Paris-Saclay supported by CNRS and Région Île-de-France, as well as it was granted access to the HPC resources of IDRIS under the allocation 2022-AD011013541 made by GENCI. This work was partially supported by the ANR Hagnodice ANR-21-CE45-0007.

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

## Appendix A. ChatGPT prompts

The main conversation with the dialogue LLM ChatGPT (OpenAI, 2022) to generate our medical prompts is the following:

- Input 1: "I want to find synonyms or text augmentations of full sentences in the context renal transplantation. Are you familiar with these concepts ?"

  Output 1: "Yes, I am familiar with the concept of finding synonyms or text augmentations of full sentences, also known as text augmentation or data augmentation. This is a technique used to artificially increase the size and diversity of a dataset by applying various natural language processing (NLP) techniques such as synonym replacement, random insertion, deletion, and substitution of words. In the context of renal transplantation, it could be used to expand the number of examples of medical reports or other texts related to renal transplantation by generating similar but slightly different versions of the original text."

- Input 2: "The full sentence template is "The age of the donor is {age}. The glomerular filtration rate (GFR) of the patient is {gfr} at {date} follow-up exam. And the creatinine levels variation were {adj}" where {age}, {gfr}, {date}, {adj} can be changed by me but not by you, do you understand ?"

  Output 2: "Yes, I understand. You have provided a sentence template with placeholders for age, glomerular filtration rate (GFR), date, and an adjective that can be changed by you. And you are looking for text augmentation to generate similar but slightly different versions of the original text with the help of natural language processing(NLP) techniques such as synonym replacement, random insertion, deletion, and substitution of words."

- Input 3: "Provide full sentence synonyms of the sentences I provided, you are allowed to give more context information and not especially respect the exact number of sentences."

  Output 3: "Here are some examples of synonyms or text augmentations of the sentence template you provided:

  1. "The age of the kidney donor is {age}. The patient's glomerular filtration rate was measured to be {gfr} at the {date} follow-up appointment. The creatinine levels showed {adj} variation."
  2. "The individual who provided the kidney for transplant was {age} years old. The recipient's GFR was determined to be {gfr} during the {date} check-up. Creatinine levels fluctuated {adj}."
  3. "At the time of transplant, the donor was {age} years of age. The patient's GFR was evaluated as {gfr} at the {date} follow-up examination. Creatinine levels demonstrated {adj} changes."
  4. "The age of the individual who donated the kidney was {age}. The GFR of the transplant recipient was {gfr} at the {date} examination. There were {adj} variations in creatinine levels."

5. "The kidney donor was {age} years old. The GFR of the transplant patient was measured as {gfr} at the {date} follow-up visit. The creatinine levels showed {adj} fluctuations."

6. "The donor's age was {age} at the time of transplant. The patient's glomerular filtration rate was found to be {gfr} during the {date} appointment. Creatinine levels were {adj}."

Please note that the sentences are not exact synonyms, but variations that convey similar information, and that it's possible to generate a lot more variations by changing the words, phrases or structures of the sentences."

- Input 4: "Give ten more synonym examples of the provided sentence and you can generate more variations as you proposed."

Output 4: "Sure, here are ten more examples of text augmentations of the sentence template you provided:

1. "The kidney donor was {age} years of age. The patient's GFR was found to be {gfr} at the {date} check-up. Creatinine levels displayed {adj} fluctuations."

2. "At the time of transplant, the age of the kidney donor was {age}. The patient's glomerular filtration rate was {gfr} at the {date} follow-up. Creatinine levels varied {adj}."

3. "The individual who provided the kidney for transplant was {age} years old. The GFR of the recipient was determined to be {gfr} during the {date} examination. There were {adj} changes in creatinine levels."

4. "The age of the kidney donor was {age}. The GFR of the transplant recipient was {gfr} at the {date} follow-up visit. Creatinine levels fluctuated {adj}."

5. "The individual who donated the kidney was {age} years of age. The patient's glomerular filtration rate was measured as {gfr} at the {date} check-up. The creatinine levels demonstrated {adj} variations."

6. "At the time of transplant, the donor's age was {age}. The GFR of the patient was evaluated as {gfr} during the {date} follow-up. The creatinine levels showed {adj} fluctuations."

7. "The kidney donor was {age} years old at the time of transplant. The patient's glomerular filtration rate was {gfr} at the {date} examination. Creatinine levels varied {adj}."

8. "The individual who provided the kidney for transplant was {age} years of age. The recipient's GFR was determined to be {gfr} at the {date} follow-up appointment. The creatinine levels displayed {adj} changes."

9. "At the time of transplant, the age of the kidney donor was {age}. The GFR of the transplant patient was {gfr} during the {date} check-up. Creatinine levels fluctuated {adj}."

10. "The individual who donated the kidney was {age} years old. The glomerular filtration rate of the transplant recipient was {gfr} at the {date} examination. The creatinine levels showed {adj} variation."

Please note that the above sentences are not exact synonyms, but variations that convey similar information, and that there are many more variations that can be generated by changing the words, phrases or structures of the sentences."

## Appendix B. Freezing the text encoder

For the text encoder, we used the BERT (Devlin et al., 2019) architecture initialized with the Bio+Clinical BERT (Alsentzer et al., 2019) model pretrained on the MIMIC clinical notes. BERT is based on the transformer (Vaswani et al., 2017) architecture and comprises 12 transformer blocks. Our main models were obtained by freezing the first 11 layers of the Bio+Clinical BERT model, fine-tuning only the last layer of the transformer with our contrastive task.

Benefiting from a dataset of 400 million (image, text) pairs collected from the internet, Radford et al. (2021) trained both their image and text encoder from scratch. While we used the same initialization as Zhang et al. (2020), they froze their text encoder's first half (6 layers). In recent NLP work, Lu et al. (2022) suggested only finetuning normalization layers (LN) in the transformer blocks, without finetuning the self-attention and feedforward layers of the residual blocks. Table 3 reports the ablation results using this latest strategy, denoted as not LN, and gradually freezing the 6, 9, and 11 first layers of the text encoder. The ablation was evaluated using two sets of weak annotations from our proposed method, first, the GFR and date, denoted as MEDIMP A, and second, the GFR, Exam, and Donor's Age, denoted as MEDIMP B. We observe that freezing the first 11 layers gives us the best performances, which is the strategy we used for our approach.

Table 3: **Ablation results on the way of freezing the text encoder.** We report F1 score (F1), and ROC AUC (AUC). **Bold**, Underlined indicates the top **1**, 2 performing combinations, respectively.

| Method | Freezing $E_\mathbf{T}$ | 2 years | | 3 years | | 4 years | | Mean | |
|---|---|---|---|---|---|---|---|---|---|
| | | AUC | F1 | AUC | F1 | AUC | F1 | AUC | F1 |
| MEDIMP A | First 11 | 81.1 | **89.4** | 81.9 | 80.0 | 74.8 | 84.4 | 79.2 | **84.6** |
| MEDIMP A | First 9 | 74.8 | 75.7 | 81.9 | 81.0 | 76.5 | 68.6 | 77.7 | 75.1 |
| MEDIMP A | First 6 | 74.2 | 74.4 | 70.0 | 82.1 | 83.5 | 68.6 | 75.9 | 75.0 |
| MEDIMP A | not LN | 73.5 | 70.6 | 77.5 | 80.0 | 73.0 | 71.8 | 74.7 | 74.1 |
| MEDIMP B | First 11 | 76.9 | 73.2 | **86.3** | **85.7** | 74.8 | **90.2** | 79.3 | 83.0 |
| MEDIMP B | First 9 | **83.7** | 64.5 | 78.1 | 82.9 | 75.7 | 64.7 | 79.2 | 70.7 |
| MEDIMP B | First 6 | 75.5 | 70.6 | 84.4 | 80.0 | **84.4** | 82.9 | **81.4** | 77.8 |
| MEDIMP B | not LN | 66.7 | 81.0 | 79.4 | 80.9 | 60.9 | 85.7 | 69.0 | 82.5 |

## Appendix C. Data preprocessing & augmentations

**Data preprocessing.** The DCE MRI volumes sized $512 \times 512 \times [64-88]$ voxels included spacing ranging in $[0.78-0.94] \times [0.78-0.94] \times [1.9-2.5]$ mm. All volumes were cropped around the transplant using an automatic and unsupervised method for selecting the region of interest and reducing dimensionality (Milecki et al., 2021). Intensity normalization was executed to each volume independently by applying standard normalization, clipping values to $[-5, 5]$, and rescaling linearly to $[0, 1]$.

**Image augmentations.** For the image data, we used data augmentation with the sequential application with each a 0.5 probability of:

- horizontal flipping;

- random affine transformation;

- random Gaussian blur ($\sigma \in [0, 0.5]$);

- random Gaussian noise ($\sigma \in [0, 0.05]$);

- random contrast perturbation ($\log \gamma \in [-0.3, 0.3]$);

using TorchIO python library (Pérez-García et al., 2021).

**Training hyperparameters.** The temperature parameter $\tau$ was initialized to 0.07, incorporated into the model as a learnable parameter, and clipped to prevent scaling the logits by more than 100, following the recommendations of CLIP training. In our experiments, we use the Adam (Kingma and Ba, 2015) optimizer with decoupled weight decay regularization (Loshchilov and Hutter, 2017) of 0.02 with a starting learning rate of $5e^{-5}$ following a cosine schedule and preceded by a linear warm-up of 40 epochs. The batch size was set to 88 and the model trained for 200 epochs with mixed-precision on 4 NVIDIA Tesla V100 GPU using Pytorch (Paszke et al., 2019).

## Appendix D. Cross-validation results

We evaluated our representations on the downstream task of kidney function prediction 2 years post-transplantation proposed by Milecki et al. (2022). We performed the task on two more prediction dates, namely 3 and 4 years post-transplantation, to better highlight the significance of our proposed approach. Nevertheless, following the instruction of Milecki et al. (2022), we also performed 10-fold cross-validation on the training set. We report below those cross-validation results (ROC AUC, F1 as mean $\pm$ standard deviation) for our validation set, for the best combination (denoted with $\star$) of CosEmbLoss (Milecki et al., 2022), CosEmbLoss++, and our proposed MEDIMP, for the three different tasks. While demonstrating similar cross-validation results over the mean of the three prediction tasks, MEDIMP enables lower variation on the validation sets.

Table 4: **Cross-validation results.** We report F1 score (F1), and ROC AUC (AUC) as mean $\pm$ standard deviation for the best combinations of CosEmbLoss, CosEmbLoss++ and our proposed MEDIMP, denoted with $\star$.

| Method | 2 years | | 3 years | | 4 years | | Mean | |
|---|---|---|---|---|---|---|---|---|
| Validation set | AUC | F1 | AUC | F1 | AUC | F1 | AUC | F1 |
| CosEmbLoss$^\star$ | $93.3\pm12.0$ | $86.4\pm12.2$ | $81.7\pm15.9$ | $74.4\pm23.3$ | $84.5\pm16.5$ | $64.6\pm26.9$ | $86.5\pm14.8$ | $75.1\pm20.8$ |
| CosEmbLoss++$^\star$ | $91.7\pm14.6$ | $88.8\pm10.4$ | $84.1\pm14.3$ | $71.4\pm20.5$ | $83.3\pm13.9$ | $69.1\pm27.3$ | $86.4\pm14.3$ | $76.4\pm19.4$ |
| MEDIMP$^\star$ | $89.3\pm11.4$ | $80.1\pm11.8$ | $87.5\pm4.1$ | $81.7\pm6.5$ | $81.4\pm13.9$ | $71.9\pm29.6$ | $86.1\pm9.1$ | $77.9\pm16.0$ |

## Appendix E. Textual data generation

We compare the proposed approach for medical prompt generation with two other approaches that produce text annotations. The first one is noted as "Manual" and comprises all the templates indicated by the medical experts, corresponding to only one sentence per variable of interest. Note that this is the base of our proposed medical prompting without using the LLM augmentation method. The second one uses an existing NLP model, T5 (Raffel et al., 2020), to produce sentences from structural data. We train the T5 model on the WebNLG 2020 data (Gardent et al., 2017) and infer it on our data to generate text, denoted as "T5 WebNLG". We observe that the "Manual" approach lacks diversity in the textual data, as no text augmentations are performed for this straightforward process. "T5 WebNLG" offers more variability in words used, but the structure of the sentences remains similar and straightforward. Moreover, some incorrect generations occur, e.g., "The age of the donor" is replaced by "The age of the patient". Such errors introduce anomalies in the data, a highly sensitive issue in such a medical context.

- "Manual" – from one sentence template:
  "The age of the donor is low. The glomerular filtration rate (GFR) of the patient is high at one month follow-up exam. And the creatinine levels variation were stable.";

- "T5 WebNLG" – pretraining a model to generate textual data from structural data:

  - correct generation example: "The age of the donor is high. The glomerular filtration rate is medium. The creatinine levels of a patient are unstable.";

  - incorrect generation example: "The age of the patient was low. The glomerular filtration rate of GFR is an extrem low rate. The creatinine levels of a patient are unstable."

- MEDIMP: see Appendix A.

