# OpenReview forum: "MEDIMP: 3D Medical Images with clinical Prompts from limited tabular data for renal transplantation"
_MIDL.io/2023/Conference — MIDL 2023 Poster_

### Official Review · Reviewer_W8SJ · 2023-02-03

**Confidence:** 5
**Preliminary Rating:** 2

**Summary:**

This work incorporates structural clinical and biological data to learn more informative representations of the renal transplant images. The proposed model, MEDIMP, is based on contrastive learning from joint text-image pairs. MEDIMP outperforms the existing SOTA in the serum creatinine (Creat) prediction task. However, the technical novelty of this work is rather weak. Please see question 3 for more details.

**Strengths:**

- The proposed method outperforms the existing SOTA.
- Ablation studies are provided. The proposed prompt generation method is compared to a T5-based table-to-text-generation method.
- The paper is easy to follow.



**Weaknesses:**

The technical novelty of this work is rather weak, specifically:
- The first contribution, as claimed in the introduction, lies in the semi-automatic medical prompt generation. The templates used for prompt generation are first manually made by experts then augmented by ChatGPT. There is no technical novelty involved in this process.
- The second contribution claimed by the authors is the learning of multi-modal representations of the renal transplant images. However, the idea of contrasting image-text pairs for better image embedding learning is not new [Zhang et al. 2020; Radford et al., 2021] and the model architecture [Devlin et al., 2018], the loss function [Zhang et al. 2020], as well as the initial parameters of the encoders [Alsentzer et al., 2019; Radford et al., 2021] of this work all come from the previous works.



**Deanonymize Review:**

no

**Paper Type:**

methodological development

**Questions To Address In The Rebuttal:**

My main concern lies in the lacking of technical novelty of this work, as detailed under question 3.  Please clarify if there is anything technical involved in the proposed prompt generation process. In addition, please clarify the differences between the proposed MEDIMP and the previous works, such as Zhang et al. (2020).

---

### Official Review · Reviewer_ibVr · 2023-02-05

**Confidence:** 5
**Preliminary Rating:** 4
**Recommendation:** Poster

**Summary:**

This paper presents MEDIMP, medical images and prompts, a model to learn meaningful multi-modal data representation for renal transplant DCE MRI by incorporating structural clinical and biological data after translating them into text prompts. The idea stems from LLMs. The goal is to learn meaningful manifolds for this particular clinical problem. This is quite an interesting approach, seems valuable tool in practical clinics.

**Strengths:**

-- a new idea, although the core is LLMs already, to conduct representation learning for multimodal data.
-- the clinical problem is an important one, predicting patients with x number of years of prognosis is valuable.
-- combining imaging and non-imaging data is towards a practical clinical case.
-- contrastive learning make it easier to enhance representation learning
-- promising results are obtained
-- Comparisons are made with manual and T5 WebNLG methods

**Weaknesses:**

-- comparisons are not throughout, limited.
-- it does not seem there is a n-fold cross validation done, the data set selection bias is potential here
-- figure 2 looks nice, but what does it show? those color mean what? what is D15. aug, D30, M3...?


**Deanonymize Review:**

no

**Detailed Comments:**

-- figure captions should go under the figure while table captions can stay above


**Paper Type:**

validation/application paper

**Questions To Address In The Rebuttal:**

I do not have many questions as the paper presents something new, and newer, and the clinical significance is high. However, I find the evaluation part a bit weak, because only 28 patients were used without cross validation. This is prone to data selection bias. Also, it is not clear why F1 is being increased across years while AUC is decreasing ?
is there any preprocessing for the MRI scans?
kidney regions look to be cropped manually, right? how big the ROIs were? standardized?

---

### Official Review · Reviewer_EShF · 2023-02-06

**Confidence:** 4
**Preliminary Rating:** 4
**Recommendation:** Poster

**Summary:**

This manuscript presented MEDIMP (Medical Images and Prompts), a model that aims to learn multi-modal representations of renal transplant Dynamic Contrast-Enhanced Magnetic Resonance Imaging (DCE MRI) by incorporating structural clinical and biological data. The model is based on contrastive learning from joint text-image paired embeddings, where the medical prompts are generated from paired clinical and biological data and augmented with automatic textual data from Large Language Models (LLMs) to provide supplementary information to pretrain image encoders. The multi-modal representations combine imaging and text embeddings based on attributes commonly used in clinical practice and complementary to imaging data. The goal of MEDIMP is to use the learned manifold of renal transplant DCE MRI to predict the transplant or patient status 2, 3, and 4 years after the transplant.

**Strengths:**

1) The paper's writing and organization are satisfactory, but it could benefit from more revisions.

2) The incorporation of structural clinical and biological data appears to hold potential for improving representation learning.

3) Discussion of the paper's limitations and weaknesses appears to be beneficial.



**Weaknesses:**

1) The paper's technical novelty appears to be limited, as using two encoders for image and text and then incorporating their representations in a single framework has been investigated in previous studies and may not be considered a significant technical advancement.

2) The lack of an essential ablation study raises concerns about the proposed method's efficacy. The use of pre-trained weights from CLIP (Radford et al., 2021) and Bio+Clinical BERT (Alsentzer et al., 2019) for the image and text encoders, respectively, raises concerns about the proposed method's contribution; thus, it is unclear what benefit the proposed method provides beyond these well-known pre-trained weights.

3) The experimental design of the paper may not provide adequate support for the proposed method's generalizability and effectiveness in a clinical setting. Based on the paper's limited results, the authors' claim that "extensive experiments and comparisons with other representation learning methods prove the effectiveness of MEDIMP, giving new directions toward medical prompts" may be considered an over-claim.

4) The use of ChatGPT in this study raises serious concerns about the proposed method's clinical reliability. ChatGPT is still in its infancy, and its use in clinical practice may raise concerns about its reliability.

5) The related work section is in need of substantial revision as numerous recent and highly relevant studies have been overlooked. For instance, in the area of self-supervised learning, the authors only mention a few works (Taleb et al., 2020 and Azizi et al., 2021) despite the presence of numerous recent publications in this field within the realm of medical imaging.


**Deanonymize Review:**

no

**Paper Type:**

methodological development

**Questions To Address In The Rebuttal:**

 Firstly, the authors must conduct a comprehensive review of recent methods that employ both image and text encoders, related to this study and clearly explain how their proposed method builds upon these. In doing so, the related work section should also be revised accordingly. Moreover, the authors must provide a clear and detailed explanation of the trustworthiness and reliability of using ChatGPT in a clinical setting, and how it can produce dependable outputs for their proposed method. Lastly, the remaining points raised as weaknesses must be addressed during the rebuttal stage.

---

### Meta-Review · Area_Chair_2K58 · 2023-02-24

**Recommendation:** Accept (Poster)
**Confidence:** 3

**Metareview:**


While it may seem quirky at first to convert tabular data to text, this work is a very nice example of exploiting LLMs for text augmentation in a way that is typically not explored, and that improves performance. Further, it offers a universal way of exploiting contrastive learning between image and text. This is very relevant to the hospital context, where virtually every patient with an image will also have medical reports of some kind (from labs, radiology etc) which are rarely exploited in an ML for healthcare context. While I agree with reviewers that text + image embeddings has been done in various guises (and so technical novelty is limited), this paper offers an interesting twist by starting from tabular data.

I also note that the authors have engaged thoughtfully with the reviews.

Pros:
- Very wide applicability
- Nice set of ablation studies

Cons:
- Mostly empirical justification (but if it works, it works!)
- Limited technical novelty


Note - authors provided new PDF with changes in another color - this is useful for reviewing but they should be reminded to submit a final one with all text in black.